



# Effective densities of soot particles and their relationships with the mixing state at an urban site of the Beijing mega-city in the winter of 2018

Hang LIU[1,2], Xiaole PAN[1], Yu WU[3], Dawei Wang[1], Yu TIAN[1,2], Xiaoyong LIU[1,4], Lu LEI[1,2], Yele SUN[1,2,4], Pingqing FU[5], Zifa WANG[1,2,4]

[1] State Key Laboratory of Atmospheric Boundary Layer Physics and Atmospheric Chemistry, Institute of Atmospheric Physics, Chinese Academy of Sciences, Beijing, 100029, China

[2] University of Chinese Academy of Sciences, Beijing, 100049, China

[3] State Key Laboratory of Remote Sensing Science, Institute of Remote Sensing and Digital Earth, Chinese Academy of Sciences, No. 20 Datun Road, Beijing 100101, China

[4] Center for Excellence in Regional Atmospheric Environment, Chinese Academy of Science, Xiamen, 361021, China

[5] Institute of Surface-Earth System Science, Tianjin University, Tianjin 300072, China

Correspondence to: Xiaole PAN (panxiaole@mail.iap.ac.cn)

**Abstract**

The effective density ($\rho_{eff}$) of refractory black carbon (rBC) is a key parameter relevant to their mixing state that imposes great uncertainty when evaluating the direct radiation forcing effect. In this study, a novel tandem DMA-CPMA-SP2 system was used to investigate the relationship between the effective density ($\rho_{eff}$) and the mixing state of rBC particles during the winter of 2018 in the Beijing mega-city. During the experiment, aerosols with a known mobility diameter ($D_{mob}$) and known $\rho_{eff}$ values (0.8, 1.0, 1.2, 1.4, 1.6, and 1.8 g/cm$^3$) were selected and measured by the SP2 to obtain their corresponding mixing states. The results showed that the $\rho_{eff}$ well represented the morphological variation in rBC-containing particles. The rBC-containing particles changed from an irregular and loose structure to a compact spherical structure with the increase in $\rho_{eff}$. A $\rho_{eff}$ value of 1.4 g/cm$^3$ was the morphological transition point. The morphology and $\rho_{eff}$ value of the rBC-containing particles were intrinsically related to the mass ratio of non-refractory matter to rBC ($M_R$). As the $\rho_{eff}$ values of the rBC-containing particles increased from 0.8 to 1.8 g/cm$^3$, the $M_R$ of the rBC-containing particles significantly increased from 2 up to 6-8, indicating that atmospheric aging processes were likely to lead to the reconstruction of more compact and regular particle shapes. During the observation period, the $\rho_{eff}$ of most rBC-containing particles was lower than the morphology transition point independent of the pollution conditions, suggesting that the major rBC-containing particles did not have a spherical structure. Simulation based on an aggregate model considering the morphological information of the particles demonstrated that absorption enhancement of rBC-containing particles could be overestimated by ~17% by using a core-shell model. This study highlights the strong dependence of the morphology of ambient rBC-containing particles on $\rho_{eff}$ and will be helpful for





elucidating the micro physical characteristics of rBC and reducing uncertainty in the evaluation of rBC climate effects and health risks.

## 1 Introduction

Refractory black carbon (rBC) is the major light absorbing aerosol in the atmosphere. It plays a vital role in the climate by influencing the radiative balance, cloud properties and glacier reduction (Flanner et al., 2007; Ramanathan and Carmichael, 2008; Bond et al., 2013). rBC is considered the second most important global warming factor after only carbon dioxide (Jacobson, 2001). Additionally, as a component of $PM_{2.5}$ (particulate matter with an aerodynamic diameter less than 2.5 μm), rBC has an adverse environmental effect due to degrading visibility and harming the human respiratory system (Apte et al., 2015; Lelieveld et al., 2015). The control of rBC emission is an immediate and win-win strategy to face climate and environmental challenges.

In the troposphere, rBC mixes with other components, such as organics, sulfate and nitrate, through condensation, coagulation or other complicated processes. Many studies have reported that the mixing state of rBC-containing particles greatly impacts on the absorption ability (Shiraiwa et al., 2008; Nakayama et al., 2010; Shiraiwa et al., 2010) and hygroscopicity of rBC (Zhang et al., 2008; Moteki et al., 2012; Liu et al., 2013) based on a combination of laboratory, numerical model and field measurement methods. However, debate exists among researchers. For instance, several studies (Lack et al., 2012; Wang et al., 2014; Wu et al., 2016; Wang et al., 2016) observed a large absorption enhancement of rBC caused by mixing with the coating material, whereas other studies found negligible absorption enhancement (Cappa et al., 2012; Lan et al., 2013). Liu et al. (2017) proposed that the mixing structure of rBC-containing particles depended on the proportion of coating material. When coating materials are insufficient to encapsulate rBC, they tend to attach to it and provide little absorption enhancement. Sufficient coating materials will change rBC-containing particles to a core-shell structure and significantly contribute to light absorption. Thus, the morphology of rBC-containing particles needs to be further studied to minimize the estimate of rBC's absorption enhancement effect. Moreover, the morphology of rBC-containing particles causes uncertainty in evaluation of the dose deposited in the respiratory system and thus in health risk estimations (Londahl et al., 2008; Alfoldy et al., 2009).

Laboratory inspection using transmission electron microscopy (TEM) can provide visual evidence of and information on the morphology of rBC-containing particles (China et al., 2013; Adachi and Buseck, 2013; Adachi et al., 2010). The common opinion based on TEM results is that bare rBC-containing particles adopt a fractal chain-like structure that will become more compact during the aging process (Wang et al., 2017). However, the representativeness of the TEM results remains a question. Since identifying rBC-containing particles using TEM is time-consuming work, the number of rBC-containing particles observed in one TEM study often ranges between hundreds and thousands, which is a tiny fraction of the ambient rBC-containing particles. Another way to determine the morphology of rBC-containing particles is to measure a physical index, such as the effective density ($\rho_{eff}$), shape factor ($\chi$), fractal dimension, etc. For instance, the $\rho_{eff}$ is defined as the ratio





of the particle mass ($M_p$) to the volume of its mobility equivalent sphere. The compactness of a particle can be determined by comparing the $\rho_{eff}$ with the material density (the density of particles with a solid spherical structure). For particles with the same material density, a smaller $\rho_{eff}$ indicates a looser structure.

In practice, a differential mobility analyzer (DMA), aerosol particle mass (APM) analyzer (or a centrifugal particle analyzer, CPMA) and condensation particle counter (CPC) are often integrated to obtain the $M_p$ and mobility diameter ($D_{mob}$)

simultaneously. Then, the $\rho_{eff}$ is calculated by Eq. (1):

$$\rho_{eff} = \frac{6M_p}{\pi D_{mob}{}^3} \tag{1}$$

The $\rho_{eff}$ is often used in laboratory studies to determine the morphology of rBC (Xue et al., 2009; Pagels et al., 2009; Zhang et al., 2008). The freshly emitted rBC-containing particles are characterized by a significantly lower $\rho_{eff}$ than the rBC material density of 1.8 g/cm$^3$ suggested by (Bond et al., 2013). Zhang et al. (2008) observed that the $\rho_{eff}$ of rBC-containing

particles changed from 0.56 to 1.60 g/cm$^3$ after $H_2SO_4$ condensation, indicating reconstruction of rBC during the condensation process, which was consistent with the TEM results. Further studies showed that BC reconstruction was caused by the surface tension of the coating material, which differed for various coating compositions (Xue et al., 2009; Pagels et al., 2009).

In the laboratory, normally a high concentration of rBC-containing particles is generated by a laminar diffusion burner, and

the $\rho_{eff}$ of rBC-containing particles can be reasonably studied using the DMA-CPMA-CPC system. However, investigating variations in the $\rho_{eff}$ of rBC-containing particles in the ambient atmosphere using a DMA-CPMA-CPC tandem system would be difficult because of the substantial presence of non-rBC particles. The $\rho_{eff}$ determined using this approach is only representative of the characteristic of the bulk aerosols and not the rBC-containing particles. A single particle soot photometer (SP2) is able to distinguish rBC-containing particles from non-rBC particles at a single particle resolution. In this

study, the CPC in the regular tandem DMA-CPMA-CPC system was replaced with the SP2, and the $\rho_{eff}$ of the rBC-containing particles and the non-rBC particles were detected separately. Additionally, the key parameters related to the rBC mixing state, such as the mass of the rBC core, number fraction of rBC-containing particles and optical diameter of the rBC-containing particles, were well determined through SP2 measurement. Thus, the mixing state and $\rho_{eff}$ of rBC-containing particles were obtained simultaneously using the novel tandem DMA-CPMA-SP2 system.

In this study, field measurement using a tandem DMA-CPMA-SP2 system was performed from Dec. 20, 2018, to Jan. 04, 2019, in the urban areas of Beijing to investigate the $\rho_{eff}$ of ambient rBC-containing particles. The site is located in the tower campus of the State Key Laboratory of Atmospheric Boundary Layer Physics and Atmospheric Chemistry, Institute of Atmospheric Physics (LAPC, longitude: 116.37°E; latitude: 39.97°N). A more detailed description of the site can be found in the literature (Sun et al., 2016; Pan et al., 2019). Particles with different effective densities preselected by the DMA-

CPMA were injected into the SP2. A comprehensive analysis was conducted with a focus on the relationship between the rBC-containing particle $\rho_{eff}$ and the mixing state. To the best of our knowledge, this study is the first report of the $\rho_{eff}$ of



ambient rBC-containing particles. This study will help elucidate the microphysical properties of rBC-containing particles, which can reduce uncertainty in climate and health risk effect estimations.

## 2 Methods

### 2.1 Instruments

#### 2.1.1 Single particle soot photometer

The detailed principal of the SP2 (Droplet Measurement Technology, Inc., Boulder, CO, USA) has been reported in the literature (Shiraiwa et al., 2008; Moteki and Kondo, 2007). Briefly, due to the unique absorption ability of rBC, each single rBC-containing particle will absorb the high intensity laser (1064 nm, TEM00 mode) produced by the SP2. Then, the rBC is 105 heated to the boiling point and emits incandescence. The peak incandescence intensity is nearly linearly correlated with the rBC mass. By detecting the incandescence, the rBC mass in each rBC-containing particle can be determined. The scattering signal of each particle is obtained simultaneously by the SP2. Particles that only have a scattering signal are identified as non-rBC particles, whereas particles with concurrence of incandescence and a scattering signal are identified as rBC-containing particles.

The SP2 was calibrated using Aquadag aerosols (lot 9627) and a polystyrene latex sphere (PSL, Nanosphere Size Standards, Duke Scientific Corp., USA) with sizes of 203 nm (lot 185856), 303 nm (lot 189903), and 400 nm (lot 189904). Because the SP2 is more sensitive to Aquadag than ambient rBC (Laborde et al., 2013), the incandescent signal was corrected by scaling a factor of 0.75 in the ambient measurement. The total uncertainty of the rBC mass measured by the SP2 was estimated to be 30%.

### 2.2 Tandem system

The tandem system in this study included a DMA (model 3085A, TSI Inc., USA), CPMA (Cambustion Ltd.), condensation particle counter (CPC, model 3775, TSI Inc., USA) and SP2. A schematic diagram of the measurement system is provided in Fig. S1.

The CPMA was used to select particles with a known mass based on a specific charge-to-mass ratio by imposing opposite 120 centrifugal and electric forces on the charged aerosols inside (Olfert and Collings, 2005). The DMA was used to select particles with a known mobility diameter ($D_{mob}$) based on the particles' electromobility. The tandem DMA-CPMA system was capable of selecting particles with a known $\rho_{eff}$. The reliability of the DMA-CPMA tandem system was tested using PSL particles ($\rho_{eff}$: 1.05 g/cm$^3$). In general, the tandem system overestimated the $\rho_{eff}$ by 5% with a mode $\rho_{eff}$ value of 1.10 g/cm$^3$, as shown in Fig. S2. The multiple charged influences (Fig. S2) were negligible.

Particles with known effective densities preselected by the DMA-CPMA system were injected into the SP2 to obtain information on the corresponding BC. In practice, the mobility diameter selected by the DMA was set at a constant value of



240 nm. The setpoints of the CPMA were 5.79, 7.24, 8.69, 10.13, 11.58, and 13.03 fg, which corresponded to a $\rho_{eff}$ of 0.8, 1.0, 1.2, 1.4, 1.6, and 1.8 g/cm³, respectively. Each CPMA setpoint was held for 10 min, and the duration of a whole scan turn of the six setpoints was 1 hour. In this study, the sum of particle numbers over ten minutes was used to present the temporal variation of particles with different $\rho_{eff}$ values.

### 2.3 Data analysis

### 2.3.1 Determination of the bulk effective density

The number concentration of particles with six different target $\rho_{eff}$ values were measured consecutively on 10-minutes basis. Thus, a distribution of different effective densities could be obtained every hour. Previous studies (Qiao et al., 2018; Momenimovahed and Olfert, 2015) often used a log-normal or Gaussian function to fit the $\rho_{eff}$ distribution. The $\rho_{eff}$ of the bulk aerosols was determined to be the peak location of the fit function. Due to the limited $\rho_{eff}$ measurement points in this study, the bulk aerosol density was calculated using a simple method as shown in Eq. 2. $\rho_i$ denotes the $\rho_{eff}$ with the maximum particle number in one hour, and $N_i$ denotes the number of particles with $\rho_i$. $\rho_{i-1}$ and $\rho_{i+1}$ denote the adjacent effective density set points of $\rho_i$.

$$\rho_{bulk} = \frac{\rho_i * N_i + \rho_{i-1} * N_{i-1} + \rho_{i+1} * N_{i+1}}{N_i + N_{i-1} + N_{i+1}} \tag{2}$$

We tested this method to calculate the $\rho_{eff}$ of PSL, as shown in Fig. S3. The $\rho_{eff}$ determined using this approach was 1.09 g/cm³, which was very close to the given density of 1.05 g/cm³.

### 2.3.2 Determination of the optical diameter

For non-rBC particles, the scattering cross-sections are proportional to the peak scattering intensity measured by the SP2. The optical diameter ($D_{opt}$) was calculated through the Mie theory with a refractive index of 1.48 and assumption of a spherical structure. Since rBC-containing particles evaporate in the laser beam, leading to a decrease in the scattering cross-section, a leading edge only (LEO) fit (Gao et al., 2007; Liu et al., 2014; Pan et al., 2017) method was used to retrieve the undisturbed peak scattering intensity. With an assumption of a core-shell structure, coating refractive index of 1.48 and BC refractive indices of 2.26-1.26i (Taylor et al., 2015), the $D_{opt}$ of the rBC-containing particles can also be calculated based on the Mie-scattering theory.

### 2.3.3 Determination of the shape factor and void fraction

The shape factor $\chi$ is an applicable parameter describing irregularity of a particle. When $\chi$ is equal to 1, the particle is in a regular spherical structure, whereas a larger $\chi$ indicates that the particle is more irregular. The $\chi$ of the rBC-containing particles was calculated using the following equation (Zhang et al., 2016) in this study:

$$\chi = \frac{D_{mob} \times C_c(D_{mev})}{D_{mev} \times C_c(D_{mob})} \tag{3}$$



where $D_{\text{mob}}$ is the mobility diameter, $D_{\text{mev}}$ is the mass equivalent diameter, and $C_C$ is the Cunningham slip correction factor (Decarlo et al., 2004).

The void volume ratio ($R_{\text{void}}$) was also used to represent the compactness of rBC-containing particles in this study. The $R_{\text{void}}$ is 0 for particles with an ideal solid spherical and increases when the structure loosens. The $R_{\text{void}}$ was calculated by Eq (4):

$$R_{void} = 1 - \frac{D_{me}^3}{D_m^3} \tag{4}$$

### 2.3.4 Determination of the coating thickness

The mass ratio ($M_R$) of coating to rBC core was used to represent the coating thickness in this study. The mass of the rBC-containing particle ($M_p$) was directly measured by CPMA and the mass of rBC core ($M_{\text{rBC}}$) was measured by SP2. Then, the $M_R$ was calculated by Eq (5):

$$M_R = \frac{M_p - M_{rBC}}{M_{rBC}} \tag{5}$$

### 3 Results

### 3.1 Constraining factors of effective density

The temporal variation in the number concentrations of rBC-containing and non-rBC particles and the mass concentration of non-refractory PM$_{2.5}$ (NR-PM$_{2.5}$) measured by a time-of-flight aerosol chemical speciation monitor (ToF-ACSM) are shown
in Fig. 1. Four pollution events and one lasting clean episode were observed during the observation period and were denoted EP 1-5. EP 3 was defined as the clean episode with a PM$_{2.5}$ mass concentration less than 10 μg/cm$^3$. The PM$_{2.5}$ mass concentration was higher than 50 μg/cm$^3$ during the other four pollution episodes. The backward trajectories of the five episodes are illustrated in Fig. S4. Beijing was majorly affected by the local air mass or southern polluted air mass during the pollution episodes. In contrast, the clean northwest air mass dominated in Beijing during the clean episode.

For simplicity, the effective density of non-rBC and rBC-containing particles were called $\rho_{\text{non-rBC}}$ and $\rho_{\text{rBC}}$ separately. Whereas the effective density of bulk non-rBC and rBC-containing particles calculated by equation (2) was called $\rho_{\text{non-rBC,bulk}}$ and $\rho_{\text{rBC,bulk}}$. The bulk effective density reflects the number distribution of particles with different effective density. For example, the number fractions of non-rBC particles with lower $\rho_{\text{non-rBC}}$ (0.8-1.2 g/cm$^3$) were ~70% during EP 2 and EP 4 and this value was significantly lower (~20%) during EP 1 and EP 5. Correspondingly, the $\rho_{\text{non-rBC,bulk}}$ was calculated to be 1.18
and 1.20 g/cm$^3$ in EP 2 and EP 4 lower than 1.43 and 1.40 g/cm$^3$ in EP 1 and EP 5. The variation in $\rho_{\text{non-rBC}}$ was mainly caused by different non-rBC compositions in different cases, since the $\rho_{\text{eff}}$ of different compositions varies. The $\rho_{\text{eff}}$ value of (NH$_4$)$_2$SO$_4$ and NH$_4$NO$_3$ particles was 1.75 g/cm$^3$ (Qiao et al., 2018), whereas those of organics depended on their compositions and usually were between 0.64 and 1.49 g/cm$^3$ (Malloy et al., 2009; Hallquist et al., 2009; Bahreini et al., 2005; Turpin and Lim, 2001). Turpin and Lim (2001) suggested an overall $\rho_{\text{eff}}$ of 1.2 g/cm$^3$ for organic aerosols in Los Angeles,



185 and Hallquist et al. (2009) recommended a $\rho_{eff}$ of 1.4 g/cm$^3$ for secondary organic aerosols in the absence of direct measurement. In general, the $\rho_{eff}$ values of organics are always lower than those of inorganic compounds. The lower $\rho_{non-rBC}$ may indicate a higher mass fraction of organic compounds in the non-rBC particles. In fact, although the composition may slightly differ between NR-PM$_{2.5}$ and particles with $D_{mob}$=240 nm as observed in this study, the higher organic fractions in NR-PM$_{2.5}$ in EP 2 and EP 4 (66% and 59%) may indicate an organic dominant pollution environment and thus a higher

190 organic fraction in particles with $D_{mob}$=240 nm consistent with the lower $\rho_{non-rBC,bulk}$ in these two episodes. Furthermore, the relationship between effective density and organic fraction was assessed throughout the observation period, as shown in Fig. S5. The $\rho_{non-rBC,bulk}$ value was apparently low in the high organic fraction environment during the whole observation period. The $\rho_{rBC,bulk}$ was also lower in the more organic fraction condition similar to that of the non-rBC particles, which was mostly due to composition of coating matters. In this study we presumed that, first, a lower $\rho_{non-rBC,bulk}$ value means a higher organic

195 fraction in the non-rBC particles. Second, composition of non-rBC particles and the coatings of rBC-containing particles were the same.

 To better understand the morphological impacts on $\rho_{eff}$, the optical diameters ($D_{opt}$) of non-rBC and rBC-containing particles were compared to that of $D_{mob}$, as shown in Fig. 2. The particle shapes are spherical if $D_{mob}$ is the same as $D_{opt}$, since the structure is assumed to be spherical in the $D_{opt}$ calculation through Mie-theory. For non-rBC particles with a $\rho_{non-rBC}$ lager

200 than 1.4 g/cm$^3$, $D_{mob}$ and $D_{opt}$ are nearly the same. For non-rBC particles with a $\rho_{non-rBC}$ =1.0 or 1.2 g/cm$^3$, the $D_{opt}$ is slightly lower than the $D_{mob}$. The refractive indices for $(NH_4)_2SO_4$, NaCl and secondary organic aerosols have been determined to be 1.51, 1.53 and 1.44-1.5, respectively (Nakayama et al., 2010; Schnaiter et al., 2003; Toon et al., 1976). An even lower refractive index (1.42) was found for ambient non-rBC particles (Zhang et al., 2018). The constant refractive index of 1.48 may underestimate the $D_{opt}$ of non-rBC particles with lower $\rho_{non-rBC}$, since the refractive indices of organic aerosols may be

205 lower than those of inorganics. Different refractive index assumptions were used to calculate the $D_{opt}$, as denoted by the red and blue dashed lines in Fig. 2. When the variation in the refractive indices was taken into account, the $D_{opt}$ was considered to be the same as the $D_{mob}$ for non-rBC particles with a $\rho_{non-rBC}$ = 1.0 or 1.2 g/cm$^3$. For non-rBC particles with a $\rho_{non-rBC}$ of 0.8 g/cm$^3$, the lower $D_{opt}$ may be caused by the lower refractive indices for some specific compounds or the non-spherical morphology. However, the fraction of this non-rBC was negligible, as shown in Fig. 1. Thus, the non-rBC particles mostly

210 adopted a spherical structure.

 The mean $D_{opt}$ was 179, 197, and 214 nm for rBC-containing particles with a $\rho_{rBC}$= 0.8, 1.0, and 1.2 g/cm$^3$, respectively, which were significantly lower than the $D_{mob}$ values. This decrease could not be explained by variation in the refractive indices, as shown in Fig. 2(b), indicating that the morphologies of these rBC-containing particles were not spherical. The $D_{opt}$ was the same as the $D_{mob}$ when the $\rho_{rBC}$ was equal to 1.6 or 1.8 g/cm$^3$, suggesting that the particles approximate to

215 spherical structure. rBC-containing particles with a $\rho_{rBC}$ = 1.4 g/cm$^3$ were placed at the morphological transition point. The differences between the 75th and 25th percentiles of the $D_{opt}$ were larger for the rBC-containing particles than for the non-rBC particles, as denoted by the box length. This larger difference may be caused by the complex morphology of rBC-containing particles compared to that of non-rBC particles. In general, the non-rBC particles mostly had a spherical structure,





and the $\rho_{non-rBC}$ was majorly influenced by the composition. A lower fraction of organics contributed to the increase in the
$\rho_{non-rBC}$. The $\rho_{rBC}$ was controlled by the combined effect of the morphology and coating composition. A fractal structure and a more organic coating tend to decrease the $\rho_{rBC}$.

**3.2 The relationship between the morphology and effective density of rBC-containing particles**

Figure 3 depicts the variations of the $\chi$ and $R_{void}$ values as a function of $\rho_{rBC}$. $\chi$ is a physical index representing the regularity of a particle; the theoretical $\chi$ for a spherical particle is 1 regardless of the void inside, and a larger $\chi$ means a more irregular
particle (Decarlo et al., 2004). In practice, the $\chi$ of rBC-containing particles ranges from 1-4 (Table 2) mostly due to the combustion material, combustion temperature, aging degree, etc. The largest $\chi$ (1.4) observed in this study was lower than that of freshly emitted rBC-containing particles from a diesel truck ($\chi$ =2.1), methane flame ($\chi$ =1.87) and propane flame ($\chi$ =4.0) and was in the range ($\chi$ =1-2.8) of rBC-containing particles with different aging degrees (Qiu et al., 2014; Peng et al., 2016). Zhang et al. (2016) suggested that the $\rho_{rBC}$ of thinly coated rBC-containing particles was 0.3 g/cm$^3$. Direct
measurements of vehicle exhaust always obtain a $\rho_{rBC}$ of 0.3-0.5 g/cm$^3$ (Momenimovahed and Olfert, 2015). Because the lower detection limit of $\rho_{rBC}$ was set to 0.8 g/cm$^3$, fresh rBC-containing particles might not have been observed in this study. Indeed, the rBC-containing particles observed in this study were actually aged rBC-containing particles with moderate irregularity.

The $\chi$ values showed a decreasing trend with the increasing $\rho_{rBC}$, indicating a more regular shape for rBC-containing
particles with a larger $\rho_{rBC}$, which was consistent with previous studies (Qiu et al., 2014; Peng et al., 2016). When the $\rho_{rBC}$ is less than 1.4 g/cm$^3$, the $\chi$ decreases significantly with the increase in $\rho_{rBC}$. However, $\chi$ varies slowly between 1 and 1.1 when the $\rho_{rBC}$ is larger than 1.4 g/cm$^3$. A similar variation trend was also found for $R_{void}$. $R_{void}$ decreases significantly from 0.5 to 0.1 and varies slowly between 0.1 and 0 when the $\rho_{rBC}$ is larger than 1.4 g/cm$^3$. These results are similar to those from the comparison between $D_{opt}$ and $D_{mob}$; the morphology of rBC-containing particles changed from an irregular and loose
structure to a compact spherical structure with the increasing $\rho_{rBC}$. Thus, a $\rho_{rBC}$ of 1.4 g/cm$^3$ may be the morphological transition point in this study. Using a smog chamber, Peng et al. (2016) also observed a change in the $\rho_{rBC}$ from ~0.5 g/cm$^3$ to 1.4 g/cm$^3$ during the aging process and found that rBC-containing particles with a $\rho_{rBC}$ = 1.4 g/cm$^3$ had a $\chi \sim 1$.

Since the $\rho_{rBC}$ was influenced by the combined effect of the coating chemical composition and morphology, the variation in $\chi$ and $R_{void}$ as a function of $\rho_{rBC}$ were ploted in Fig. 3. We found that rBC-containing particle with a higher coating effective
density (1.5<$\rho_{non-rBC,bulk}$<1.7 g/cm$^3$) had a larger $\chi$ value and $R_{void}$ than that with 1.1<$\rho_{non-rBC,bulk}$<1.3 g/cm$^3$, especially for irregular particles (Fig. 3a). It implied of different coating composition may play a different role in the morphology reconstructing of rBC-containing particle and will be further discussed in Section 3.3. The rBC-containing particles with a lower coating effective density could reach a compact spherical structure when the $\rho_{rBC}$ was 1.2 g/cm$^3$ with an $\chi$ of 1.05 and a $R_{void}$ of 0.92, whereas the morphological transition of $\rho_{rBC}$ was higher for rBC-containing particles at a higher $\rho_{non-rBC,bulk}$
condition.





### 3.3 Mass ratio of coatings to the rBC core of rBC-containing particles with different $\rho_{eff}$ values

The mass ratio ($M_R$) of the coating to the rBC core is used to represent the coating thickness in this study. The coating thickness is an index of the aging degree of rBC-containing particles since condensation and coagulation will lead to an increase in the coating thickness during the aging process. As shown in Fig. 4, rBC-containing particles with larger $\rho_{rBC}$

values had more coating. This phenomenon explains the morphological change that occurs with an increasing $\rho_{rBC}$. The surface tension imposed by the coating was found to shrink the rBC core (Zhang et al., 2016). After coating, rBC-containing particles with larger hygroscopicity more easily obtain surface water, which enlarges the surface tension and shrinks the rBC-containing particles to a more compact structure (Zhang et al., 2008). Moreover, the coatings are able to fill the void of rBC-containing particles, resulting in a more compact structure (Pagels et al., 2009). Thus, increasing the coating makes

rBC-containing particles more compact, and the compact structure leads to a larger effective density, as observed.

Recently, studies using different methods have proven the occurrence of morphological change of rBC-containing particles with an increase in the coating thickness. Peng et al. (2016) observed that rBC-containing particles changed to a compact spherical structure when the ratio of the coating thickness to the rBC core diameter reached 0.8-1, corresponding to a $M_R$ of 4.5-6.5. By comparing the measured and modeled scattering cross-sections of rBC-containing particles, Liu et al. (2017)

found that the measured scattering cross-section agreed well with the core-shell model prediction when the $M_R$ was larger than 3, suggesting adoption of a spherical morphology by rBC-containing particles with a large $M_R$. The morphology of rBC-containing particles seems to change to be spherical at a certain $M_R$ point. In this study, the $M_R$ was nearly invariant and fluctuated between 6 and 8 when the $\rho_{eff}$ was larger than 1.4 g/cm$^3$, suggesting that the rBC-containing particles were mostly spherical in structure when the $M_R$ was larger than 6-8.

The $M_R$ was also counted in different coating composition conditions represented by $\rho_{non-rBC,bulk}$. When the $\rho_{rBC}$ was between 0.8 and 1.2 g/cm$^3$, the $M_R$ was larger at the lower $\rho_{non-rBC,bulk}$ condition, leading to a more regular shape that was consistent with the morphological characteristics shown in Fig. 3. With the increase in the $\rho_{rBC}$, the $M_R$ gradually increased and became nearly invariant. At the lower $\rho_{non-rBC,bulk}$ condition (e.g., 1.1 g/cm$^3$ <$\rho_{non-rBC,bulk}$< 1.3 g/cm$^3$), the $M_R$ reached the invariant point at the lower $\rho_{rBC}$ (1.2 g/cm$^3$) corresponding to the lower morphological transition $\rho_{rBC}$ point discussed in section 3.2.

Notably, the $M_R$ invariant point was lower (6.5) at the lower $\rho_{non-rBC,bulk}$ condition (1.1 g/cm$^3$ <$\rho_{non-rBC,bulk}$< 1.3 g/cm$^3$). This phenomenon may have two explanations. First, the coating composition may play an important role in the morphological change of rBC-containing particles. By summarizing the TEM results, Adachi et al. (2010) presumed a mixing process in which the organic first condensed and filled the void of rBC and then the sulfate aerosol coagulated on the rBC-containing particles. The different coating methods may cause different structures of rBC-containing particles. In our observation, the

rBC-containing particles with the more organic-like coating (1.1 g/cm$^3$ <$\rho_{non-rBC,bulk}$< 1.3 g/cm$^3$) more easily formed a spherical structure with a lower $M_R$. Second, Pagels et al. (2009) suggested the volume ratio rather than the mass ratio determined the restructuring efficiency of rBC. The morphological transition volume ratio was calculated to be similar at different $\rho_{non-rBC,bulk}$ conditions, since the morphological transition $M_R$ was lower at the lower $\rho_{non-rBC,bulk}$ condition.



In the real atmosphere, the variant temperature and relative humidity may also contribute to the morphological variation of
rBC-containing particles, which makes the morphological transition $M_R$ point more complicated and thus different from that
reported in previous studies. However, an agreement has been reached concerning the mechanism by which the morphology
changes with the increasing coating thickness. A $M_R$ of 7 or a volume ratio of 9 was determined to be the morphological
transition point in Beijing in winter. The morphological transition $M_R$ or volume ratio may be very useful for
parameterization in atmospheric models. Thus, more observations are needed to explore the variation and constraining
factors of the morphological transition $M_R$.

Previous studies have always used a shell/core ratio (S/C) to represent the coating thickness. The $M_R$ of rBC-containing
particles with a $\rho_{rBC}$ =0.8 g/cm$^3$ averaged 2.0, corresponding to an S/C of 1.5, and the $M_R$ of rBC-containing particles with a
$\rho_{rBC}$ =1.4 g/cm$^3$ averaged 7.0, corresponding to an S/C of 2.15. Typically, the S/C ratio of freshly emitted BC observed at
urban sites was lower than 1.2 (Liu et al., 2014; Laborde et al., 2013). In this study, rBC-containing particles with a $\rho_{rBC}$ of
0.8 g/cm$^3$ were characterized as having an irregular and loose structure with $\chi = 1.4$ and $R_{void}$=0.5. Since the $\rho_{rBC}$ increased
with the increase in the $M_R$ or S/C, the observed average S/C = 1.2 at the urban site suggested that most rBC-containing
particles in the urban site might have a $\rho_{rBC}$ lower than 0.8 g/cm$^3$ and thus a more irregular structure.

**3.4 The morphology of bulk rBC-containing particles**

The number distributions of $\rho_{rBC}$ and $\rho_{non-rBC}$ were counted to evaluate the morphological characteristics of the bulk rBC-
containing particles in the ambient environment, as shown in Fig. 5. Generally, the $\rho_{rBC}$ and $\rho_{non-rBC}$ number distributions
exhibited a mono-modal distribution except for the $\rho_{rBC}$ distribution in EP 3 (the clean period). Observations of rBC-
containing particles indicated that the coating thickness distribution always exhibited a clear bimodal pattern (Liu et al., 2014;
Wu et al., 2017). rBC-containing particles were observed with a thin coating, which was mostly attributed to local traffic
emissions, and with a thick coating, which might be the result of biomass emissions or the aging process. The aged rBC-
containing particles with a $\rho_{rBC} > 0.8$ g/cm$^3$ observed in this study may exactly correspond to the rBC-containing particles
with a thick coating and thus exhibit a unimodal pattern. We speculate that the $\rho_{rBC}$ number distribution will exhibit a
bimodal pattern if the detection limit is sufficiently low and the left peak in the expected bimodal pattern corresponds to the
rBC-containing particles with thin coatings. The $\rho_{rBC}$ distribution in EP 3 may be influenced by the thinly-coated rBC-
containing particles, which might be present in relatively higher numbers during the clean period.

The $\rho_{rBC,bulk}$ and $\rho_{non-rBC,bulk}$ were separately estimated to be 1.21 g/cm$^3$ and 1.39 g/cm$^3$, respectively, throughout the
observation period. Notably, due to the detection limit, the $\rho_{rBC,bulk}$ in this study was determined for aged rBC-containing
particles, and the true $\rho_{rBC,bulk}$ was expected to be lower if fresh rBC-containing particles were taken into consideration.
However, even for these aged rBC-containing particles, the morphology was mostly in a fractal structure, because the
$\rho_{rBC,bulk}$ (1.21 g/cm$^3$) was smaller than the morphological transition $\rho_{rBC}$ (1.40 g/cm$^3$). Wang et al. (2017) proved that only 12%
of rBC-containing particles were in an embedded structure and that 88% of rBC-containing particles were in a bare or partly



coated structure in urban sites through direct TEM observation. Our results provided evidence for the irregularity of rBC-containing particles based on assessment of more particle numbers.

The $\rho_{eff}$ was separately counted in the five episodes, as shown in Fig. 5, to investigate variation in $\rho_{eff}$ under different

pollution situations. The $\rho_{rBC,bulk}$ was 1.37, 1.01, 1.15, 1.01, and 1.21 g/cm³ during EP 1-5, and the corresponding $\rho_{non\text{-}rBC,bulk}$ was 1.43, 1.18, 1.40, 1.20, and 1.40 g/cm³, respectively. According to Fig. 3a, the morphological transition $\rho_{rBC}$ points were 1.2 g/cm³ in EP 2 and EP 4 and 1.4 g/cm³ in EP 1, EP 3, and EP 5 due to the different $\rho_{non\text{-}rBC,bulk}$ value. The $\rho_{rBC,bulk}$ in the five episodes was lower than that of the morphological transition $\rho_{rBC}$ regardless of the pollution conditions, indicating that a substantial number of rBC-containing particles were in a fractal structure even under pollution conditions. However, the

$\rho_{rBC,bulk}$ in EP3 was smaller than that in EP1 and EP5, which might suggest a more compact structure of rBC-containing particles in pollution conditions, since the morphological transition $\rho_{rBC}$ was similar during these three episodes.

The number fraction of rBC-containing particles in the total measured particles (rBC-containing and non-rBC particles) increased with the decrease in the $\rho_{eff}$, as shown in Fig. 6. rBC-containing particles only accounted for 10-20% of particles with a $\rho_{eff}$ = 1.6 g/cm³, whereas this fraction significantly increased to ~60% for particles with a $\rho_{eff}$ = 0.8 g/cm³. The data

from the five episodes all followed the same tendency, and the maximum number fraction of rBC-containing particles was reached when the $\rho_{eff}$ was equal to 0.8 g/cm³. A power function was used to fit the data and showed that the number fraction of rBC would be 100% if the $\rho_{eff}$ was less than 0.73 g/cm³. Rissler et al. (2014) observed a bimodal $\rho_{eff}$ distribution of ambient aerosols. The $\rho_{eff}$ of aerosols in the two peaks ranged separately from 0.30 - 0.80 g/cm³ and 1.28 – 1.46 g/cm³. The mass of aerosols with a $\rho_{eff}$ of 0.30 - 0.80 g/cm³ only lost 10% after being heated to 300 °C, indicating that most of these

particles were fresh rBC-containing particles, which was consistent with our inference. Thus, if the measurement site was located in an area with enough fresh BC emission, the bimodal $\rho_{eff}$ distribution of ambient aerosols was often observed (Qiao et al., 2018; Liu et al., 2015; Rissler et al., 2014), since the $\rho_{eff}$ of fresh rBC-containing particles was sufficiently small with no disturbance by non-rBC particles. However, as shown in Fig. 6, the $\rho_{eff}$ distributions of non-rBC and aged rBC-containing particles overlapped and could not be distinguished through a simple DMA-APM/CPMA-CPC system. Our study suggested

that the second peak often observed in previous $\rho_{eff}$ measurements was actually a mixture of non-rBC and aged rBC-containing particles. Rissler et al. (2014) used "dense" particles to describe particles with a $\rho_{eff}$ of 1.28 – 1.46 g/cm³. This expression might be not very accurate, since nearly 10-40% of the particles were aged rBC-containing particles with a fractal structure ($\chi$=1-1.2).

### 3.5 Optical properties of rBC-containing particles with different $\rho_{eff}$

Since the morphology of rBC-containing particles was mostly in a fractal structure as discussed above, the simple core-shell structure treatment in the atmospheric model might cause bias in the optical property estimate of rBC-containing particles. An aggregate model was established, and the optical properties were calculated by solving Maxwell's equation based on the superposition T-matrix method (Wu et al., 2018). As shown in Fig. 7(a), large discrepancies in the scattering cross-sections ($\sigma_{SC}$) between the core-shell model and measurement were found in the small $\rho_{rBC}$ range, indicating the strong impact of





morphology on optical properties. An aggregate model can better capture the $\sigma_{SC}$ characteristics than a perfect shell-core model when the $\rho_{rBC}$ is smaller than 1.4 g/cm$^3$. When the $\rho_{rBC}$ is 0.8 g/cm$^3$, the $\sigma_{SC}$ predicted by the aggregate model with an rBC core fractal dimension ($D_f$) of 2.0-2.2 agrees well with the measurement. With an increase in the $\rho_{rBC}$, the measured $\sigma_{SC}$ is consistent with the predicted value from the aggregate model obtained using the larger rBC core fractal dimension. This result may imply that the BC core becomes more compact and regular with an increase in the $\rho_{rBC}$ or coating thickness

consistent with the laboratory results (Pagels et al., 2009; Xue et al., 2009).

Figure 7(b) exhibits the estimated mass absorption cross-sections (MACs) from different models. The overestimation of the MAC using the core-shell structure averaged 16.7% compared to that of the aggregate model when the $\rho_{rBC}$ was less than 1.4 g/cm$^3$. Additionally, the measured $\sigma_{SC}$ with a $\rho_{rBC}$= 1.6 g/cm$^3$ was similar to the predicted values of models 1 and 2, indicating a near spherical structure of the rBC-containing particles. However, the MAC predicted by models 1 and 2 varied.

Although laboratory studies proved that the rBC core shrank after coating, an irregular rBC core was often observed even in thickly-coated cases in the ambient measurement (Adachi et al., 2010; Zhang et al., 2016), suggesting that model 2 might be closer to the realistic thickly-coated situation. Since model 1 overestimated the MAC by 7.4% with a $\rho_{rBC}$ = 1.6 g/cm$^3$ compared to that of model 2, the morphology of the rBC core should also be considered, even in cases with a large $\rho_{rBC}$ or thickly-coated condition. In general, the commonly observed light absorption enhancement in pollution conditions cannot

simply be attributed to the "lensing effect". The morphological change of rBC-containing particles and the BC core may also play an important role in light absorption enhancement.

## 4 Conclusion

A novel tandem DMA-CPMA-SP2 system was used to investigate the effective density of rBC-containing particles ($\rho_{rBC}$) and their relationship with the rBC mixing state in Beijing. Aerosols with the same mobility diameter (240 nm) and different

$\rho_{eff}$ values (0.8, 1.0, 1.2, 1.4, 1.6, and 1.8 g/cm$^3$) were preselected by the DMA-CPMA system and injected into the SP2 to obtain the corresponding mixing state. The results showed that the $\rho_{rBC}$ could reflect the morphology of rBC-containing particles. The dynamic shape factor of rBC-containing particles decreased from 1.4 to 1 with the increase in the $\rho_{rBC}$, indicating that the morphology of the rBC-containing particles changed from an irregular loose structure to a compact spherical structure. rBC-containing particles with $\rho_{rBC}$ values of 0.8, 1.0, and 1.2 g/cm$^3$ mostly adopted a non-spherical

structure, whereas those with $\rho_{rBC}$ values of 1.6 and 1.8 g/cm$^3$ were in a spherical structure. The $\rho_{rBC}$ = 1.4 g/cm$^3$ was determined to be the morphological transition point in this study. The mass ratio ($M_R$) of the coatings to the rBC core was calculated for rBC-containing particles with different $\rho_{rBC}$ values. The $M_R$ gradually increased from 2 to 6-8 with the increase in the $\rho_{rBC}$ when that latter measure was less than 1.4 g/cm$^3$ and stayed invariant when the $\rho_{rBC}$ was larger than 1.4 g/cm$^3$, suggesting that the increased coating thickness during the aging process was the cause of morphological changes and that the

rBC-containing particles tended to be spherical when the $M_R$ was larger than 6-8 in the winter in Beijing.



The morphological characteristics of the bulk ambient rBC-containing particles were investigated by calculating the bulk effective density of rBC-containing particles ($\rho_{rBC,bulk}$) considering the number distribution of $\rho_{rBC}$. The $\rho_{rBC,bulk}$ averaged 1.21 g/cm$^3$ during the whole observation period and was lower than the morphological transition $\rho_{rBC}$ regardless of the pollution conditions. The $\rho_{rBC,bulk}$ was overestimated due to the lower detection limit in this study (set to 0.8 g/cm$^3$), which
was larger than the $\rho_{rBC}$ of freshly emitted rBC-containing particles. However, the $\rho_{rBC,bulk}$ was still lower than the morphological transition $\rho_{rBC,}$ suggesting that the rBC-containing particles were mostly not in a core-shell structure in the ambient condition. An aggregate model considering the morphological information of rBC-containing particles was approved to better represent and to evaluate the optical properties of rBC-containing particles. Generally, the core-shell model overestimated light absorption compared to that of the aggregate model by 16.7% for rBC-containing particles with a
$\rho_{rBC}$ =0.8 -1.4 g/cm$^3$. This study revealed that a substantial number of rBC-containing particles were in an irregular structure in the ambient atmosphere and highlighted the importance of morphology for optical property estimates. A proper parameterization considering rBC-containing particle morphological changes with $M_R$ and a morphology-dependent optical model may help reduce the uncertainty in atmospheric modeling.


**Data availability**

To request the data given in this study, please contact Dr. Xiaole Pan at the Institute of Atmospheric Physics, Chinese Academy of Sciences, via email (panxiaole@mail.iap.ac.cn).

**Author contributions**

H.L, X.P designed the research; H.L, X.P, D.W, X.L, Y.T, Y.S, P.F, Z.W performed experiments; H.L, X.P, Y.T performed the data analysis; H.L, X.P wrote the paper.

**Acknowledgements**

This study was supported by the National Natural Science Foundation of China (No. 41605104).



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





**Table 1. Abbreviations and symbols used in this paper**

| Abbreviation/symbols | Full name/explanation |
|---|---|
| $\rho_{non\text{-}rBC}$ & $\rho_{non\text{-}rBC}$ | Effective density of non-rBC particles (rBC-containing particles) |
| $\rho_{non\text{-}rBC,bulk}$ & $\rho_{rBC,bulk}$ | Effective density of bulk non-rBC particles (rBC-containing particles) using the weighted average method. $\rho_{non\text{-}rBC,bulk}$ is different from $\rho_{non\text{-}rBC}$, since the number distribution of non-rBC particles with different $\rho_{non\text{-}rBC}$ values is considered to reflect the effective density of bulk non-rBC. |
| SP2 | Single particle soot photometer (DMT Technologies) |
| DMA | Differential mobility analyzer (TSI Inc.) |
| CPMA | Couette centrifugal particle mass analyzer (Cambustion, Ltd.) |
| CPC | Condensation particle counter (TSI Inc.) |
| rBC | Refractory black carbon determined by the SP2 through the laser-induced incandescence method |
| $D_{mob}$ | Mobility diameter selected by the DMA |
| $D_{opt}$ | Optical diameter derived from the SP2 scattering signal |
| $\chi$ | Dynamic shape factor of particles representing the particle regularity |
| $R_{void}$ | Void volume ratio in a particle representing the particle compactness |
| $M_R$ | The mass ratio of coatings to the rBC core representing the coating thickness |
| S/C ratio | The ratio of the diameter of rBC-containing particles to the diameter of the rBC core representing the coating thickness |
| $\sigma_{SC}$ | Scattering cross-section of rBC-containing particles |
| MAC | Mass absorption cross-section of rBC-containing particles |




**Table 2. Brief summary of the effective density and dynamic shape factor of rBC-containing particles**

| Particle type | Shape factor* | Effective density* (g/cm³) | Description | Reference |
|---|---|---|---|---|
| ambient | - | 1.42 (0.39) | The $\rho_{eff}$ distribution exhibits a bimodal pattern. The left peak is contributed by fresh soot with a $\rho_{eff}$ of 0.39 g/cm³. | (Rissler et al., 2014) |
| ambient | - | 1.50 (0.8) | Similar bimodal $\rho_{eff}$ distribution. | (Qiao et al., 2018) |
| non-rBC (dioctyl sebacate) | 1.03 | 0.931 | The shape factor is nearly 1 for non-rBC particles. | (Tavakoli and Olfert, 2014) |
| rBC-containing particles | 2.10 | 1.00 | rBC-containing particles from a diesel truck. | (Han et al., 2019) |
| | 4 | 0.18 | rBC-containing particles generated from a propane flame. | (Xue et al., 2009) |
| | 1.80 | 0.70 | Propane flame rBC-containing particles coated with glutaric acid. | |
| | 1.87 | 0.50 | rBC-containing particles generated from a methane flame. | (Tavakoli and Olfert, 2014) |
| | 1.03 - 2.79 | 1.36 - 0.25 | Diesel exhaust rBC-containing particles with different combustion temperatures; the rBC-containing particles were more compact from the lower temperature combustion condition. | (Qiu et al., 2014) |
| | 1 - 2.8 | 1.4 – 0.25 | Laboratory-generated rBC-containing particles with different aging times; the rBC-containing particles were more compact when undergoing a long aging process. | (Peng et al., 2016) |

*The effective density and shape factor counted in this table is for rBC-containing particles with a $D_{mob}=240\pm20$ nm.



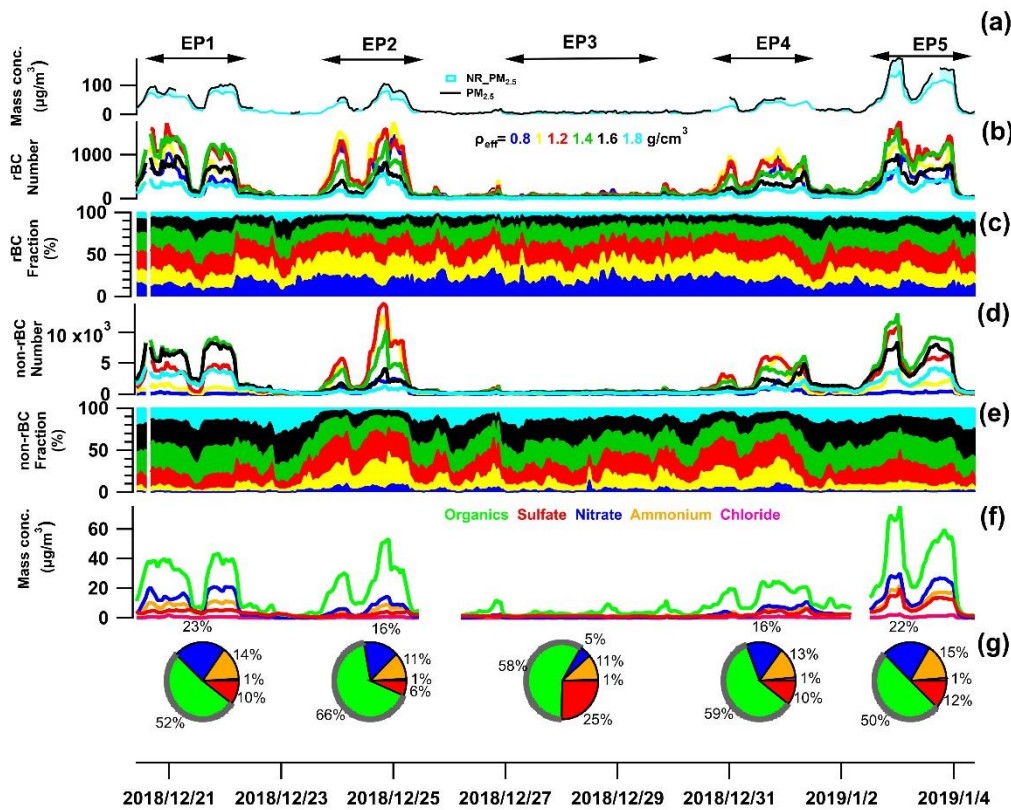


**Figure 1. Time series of (a) the PM₂.₅ mass concentration, (b) number counts of rBC-containing particles with different effective densities, (c) number fractions of rBC-containing particles with different effective densities, (d) number counts of non-rBC particles with different effective densities, (e) number fractions of non-rBC particles with different effective densities, (f) mass concentrations of aerosol species, including organics, sulfate, nitrate, ammonium, and chloride, in NR-PM₂.₅ and (g) mass fractions of aerosol species in NR-PM₂.₅ during the five episodes denoted at the top of the graph.**



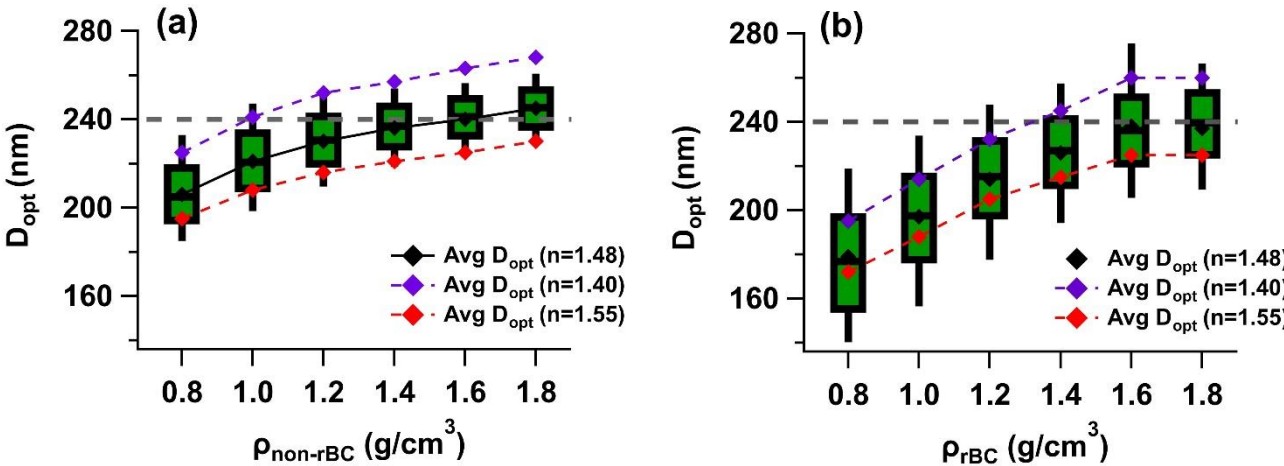

**Figure 2. The optical diameters of particles with different effective densities. (a) Non-rBC particles and (b) BC-containing particles. (The black lines in the middle signify the medians; the black markers in the middle denote the means; the upper and lower bounds of the box denote the 75th and 25th percentiles, respectively; and the upper and lower whiskers denote the 90th and 10th percentiles, respectively.) The blue and red dashed lines denote the average optical diameters from different assumptions of the refractive index. The grey dashed line denotes the mobility diameter ($D_{mob}$ = 240 nm) selected by DMA.**

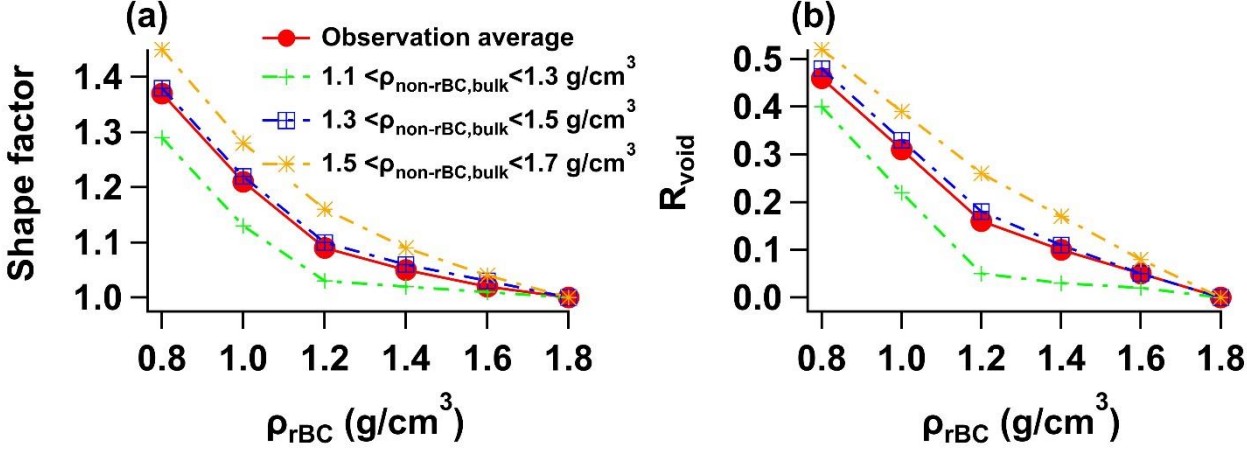

**Figure 3. The shape factor and void fraction of BC-containing particles under different effective densities.**





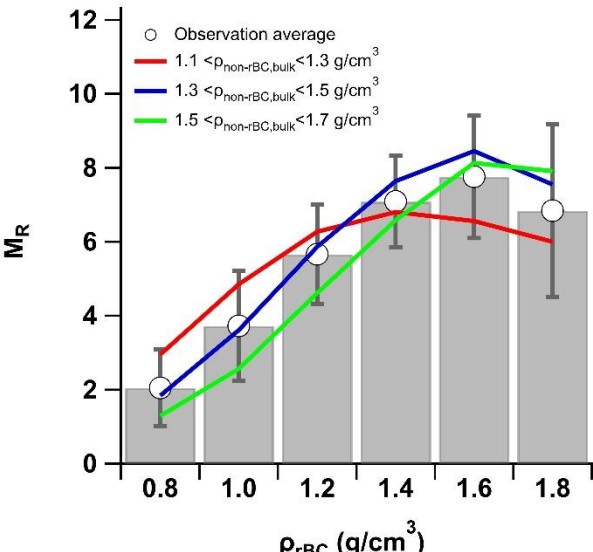

**Figure 4.** The mass ratio ($M_R$) of coatings to rBC for different $\rho_{rBC}$.






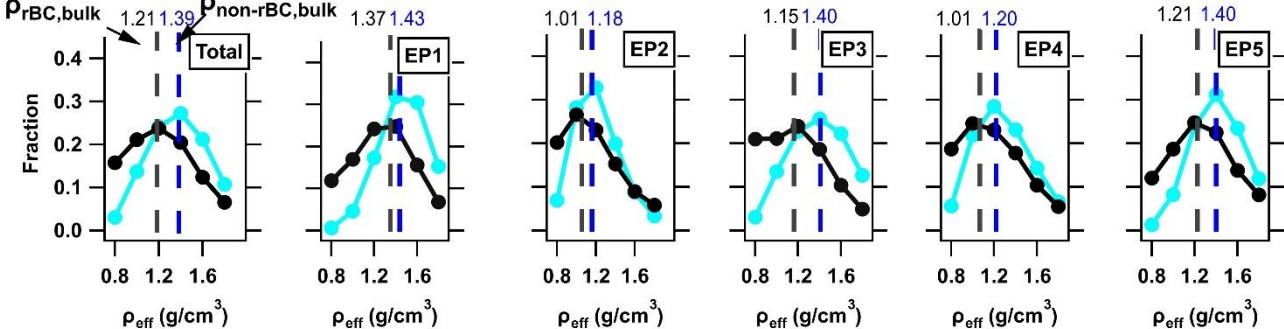

**Figure 5. Number distributions of rBC-containing particles (black lines) and non-rBC particles (blue lines) with different effective densities during different episodes. The black dashed line denotes the effective density of bulk rBC-containing particles, and the blue dashed line denotes the effective density of bulk non-rBC particles.**


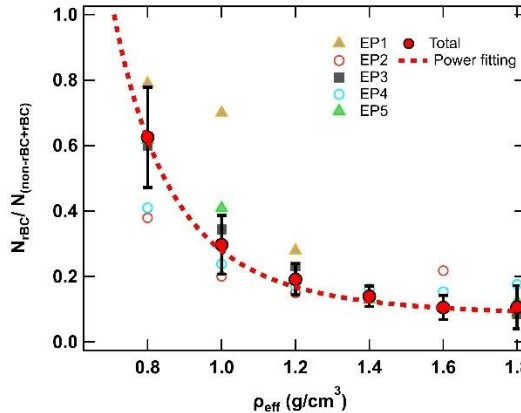

**Figure 6. Number fractions of rBC-containing particles in the total particles (rBC-containing and non-rBC particles) under different effective densities.**





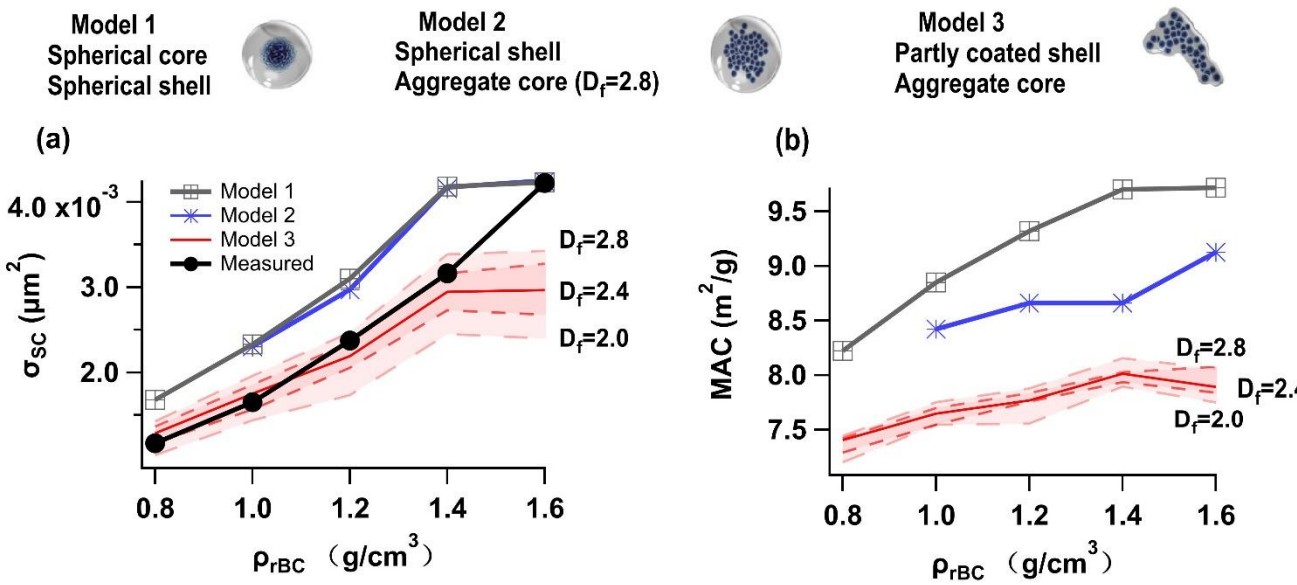


**Figure 7. (a) The scattering cross-section of BC-containing particles at the 1064 nm wavelength predicted by different models and measured by the SP2 under different effective densities. (b) The mass absorption cross-section at the 532 nm wavelength predicted by different models under different effective densities. Model 3 was used with various assumptions of the core fractal dimension ($D_f$= 2.0, 2.2, 2.4, 2.6, and 2.8) as denoted in the graph.**
