# Peer review of "Effective densities of soot particles and their relationships with the mixing state at an urban site of the Beijing mega-city in the winter of 2018"

_Atmospheric Chemistry and Physics, 2019_

## Referee Comment (RC1) · Anonymous Referee #1 · 26 Aug 2019

This manuscript describes relation among density, morphology, and mixing states of aerosol particles containing soot. On the whole, the topic of the manuscript is relevant and suitable for the scope of the "Atmospheric Chemistry and Physics. The results taken from this study deserve to be made available to the scientific community and to be exploited in terms of atmospheric aerosols and atmospheric radiation budgets.

This manuscript is basically well-written and discussed carefully. I recommend that this manuscript is published in ACP after some corrections (as listed below).

1. Measurement procedures (Section 2.2) Aerosol particles were measured in the six steps using DMA-CPMA tandem system. In analytical condition, it took 10 minutes for

[Figure]

[Figure]

each step (whole scan needs one hour). What is the response to achieve equilibrium in step change? Usually, data after equilibrium condition were used in the stepwise measurements to reduce analytical errors. In other words, data immediately after step change were not suitable for analysis.

2. Refractive index (Section 2.3.2) Refractive index of 1.48 + 0i was used for measurements of optical diameter. What is aerosol components with refractive index of 1.48 + 0i? This information is helpful for readers who are not familiar with aerosol optical properties.

3. Line 177-180 Effective density of non-rBC with 0.8-1.2 g cm-3 (brue, yellow, and red) was no ∼20 % during EPs 1 and 5. The density was 30-40 % during EPs 1 and 5 in Fig. 1.

4. Line 206-207 and Fig. 2 Figure 2 provides us very important and interesting knowledge on relation between aerosol size and effective density. Before detail discussion, influence of analytical error should be discussed. It is true that variation of refractive index can influence optical diameter. However, analytical error or analytical precision in the tandem system needs to be taken into account.

5. Line 247-250 Rvoid value (0.92) was shown in this sentence. In Figure 3, Rvoid of 0.92 seems to be too large. Is that typo?

6. Line 307-309 As shown in the text, insufficient detection limit can lead to disturb identification of detection of bi-modal pattern. Also, low analytical resolution in density may be one of the reasons.

7. Line 322-326 This difference between Eps 2 &4 and Eps 1 & 5 is very interesting in comparison of ambient aerosols. Are there any differences in meteorological conditions, air mass history, and so on? What were important or key processes to engender the differences?

8. Figures 5-7 Analytical errors (i.e., error bars) should be added in the plots.

---

## Referee Comment (RC2) · Anonymous Referee #2 · 26 Aug 2019

The manuscript is a nice piece of work using innovative combination of up-to-date methodologies to shed light on the properties of rBC in ambient air. In this respect, the study is novel in the field and brings up some interesting results that can be useful for both the modelling and experimental communities.

However, I feel myself a bit uncomfortable with the approach the authors have taken in the followings:

1) While the authors are fully aware that rBC particles in ambient air range in their effective densities from as low as 0.3 g/cm3 (see Page 8 Line 230 and references therein), they arbitrarily set the lowest limit of detection to 0.8 g/cm3, thus giving up a

sizable fraction of rBC particles in ambient air. This makes their conclusions truncated that need to be supplemented with extrapolation and to some extent speculations. Why did they do this? One might presume that by setting the mobility diameter to 240 nm they perhaps thought that native (freshly emitted) rBC particles having very low effective densities are not relevant in this range?

2) In spite of the fact that several important physical parameters used in this study are derived by subtraction and division of measured quantities that were obtained by fundamentally different techniques (e.g. equations 4 and 5), the authors pay little if any attention to (propagated) uncertainties that can be huge in these cases.

Detailed comments:

Page 2 Line 37 'glacier reduction' is an imprecise term. Reduction in what? length? volume? albedo? and how? Sea ice and snow albedo is also reduced and melt is affected directly (e.g. by albedo reduction) and indirectly (e.g. by affecting radiative balance over reflective surfaces by absorbing reflected radiation)

Page 2 Line 38 Second most important warming agent. There is no consensus yet whether methane or rBC comes second. Reference outdated, please update references and modify the statement accordingly

Page 2 Line 40 'Visibility degradation' is not a major effect of rBC, it is mostly due to scattering aerosols. It is not the human respiratory system that is directly 'harmed', but soot has many adverse health effects (including cardiovascular illnesses, cancer, and even brain damage)

Page 2 Line 44 'other complicated processes': heterogeneous chemistry, including cloud processing is of utmost importance in affecting the mixing state of submicron particles, it should be mentioned separately

Page 2 Line 54 'minimize the estimate'? minimize the error in the estimation. . .

Page 2 Line 82 'substantial presence' please rephrase

Page 4 Line 103 'principal'?

Page 4 Line 105 'emits incandescence' please rephrase

Page 5 Line 148-149 flaws in logics: here morphology and properties of BC is preset for the calculations, while the major objective of the study is to determine both

Page 6 Line 165: uncertainty should be reported here since the two parameters are determined by two principally different methods having their own inherent uncertainties

Page 6 Line 171-172 'microgram/cm3' ?

Page 6 Line 175 'density' use plural

Page 7 Line 195 this statement is definitely not true. Bulk non-rBC particles should differ from the coatings of rBC due to differences in chemistry of their formations.

Page 8 Line 243-249 The whole statement is highly speculative (see my comment immediately above)

---

## Short Comment (SC1) · 5 Sep 2019

Liu et al. use a tandem DMA-CPMA-(CPC/SP2) system to measure the effective densities of refractory black carbon (rBC) particles in the atmosphere of Beijing. The setup of DMA-CPMA-SP2 is very novel. However, this paper has obvious weakness in the data interpretation, which is sometimes rather imprecise.

I use several examples to illustrate my main concerns of the paper. 1. Comment on the representative of rBC particles in this study This paper only focuses on the rBC particles with effective density from 0.8 to 1.8 g cm-3. Actually, lots of rBC particles emitted from emission sources show a much lower effective density. Unfortunately,

they are not included in this study.

2. Comment on the calculation and definition of effective density This paper presents two methods of effective densities calculation, but I do not quite understand them. Previous studies (Qiao et al., 2018; Momenimovahed and Olfert, 2015) generally used a log-normal or Gaussian function to fit the eff distribution. The eff of the bulk aerosols was determined to be the peak location of the fit function. In this paper, however, the first calculation method (Lines 125-130) is, "Particles with known effective densities preselected by the DMA-CPMA system were injected into the SP2 to obtain information on the corresponding BC. In practice, the mobility diameter selected by the DMA was set at a constant value of 240 nm. The setpoints of the CPMA were 5.79, 7.24, 8.69, 10.13, 11.58, and 13.03 fg, which corresponded to a eff of 0.8,1.0, 1.2, 1.4, 1.6, and 1.8 g/cm3, respectively." This calculation is totally different from the calculation in the previous studies. Is no a log-normal or Gaussian function fitted? It is imprecise in theory and in practice. What is the strategy about the selection of CPMA data?

The second calculation method defines a new effective density which names the bulk aerosol density, as stated in Lines 137. In my opinion, it should be the bulk aerosol effective density. Additionally, the authors simply use the PSL to demonstrate this method, which lacks the experiments about rBC.

3. Comment on the setup of DMA-CPMA-SP2 This study uses a novel setup to characterize the effective density of rBC particles in the atmosphere, but this setup does not be verified by rBC particles produced in laboratory. I strongly suggest that it should be assessed before applying it to the field observation.

4. Comment on the determination of the shape factor In 2.3.3, author uses the equation (3) to determine the dynamic shape factor. In the equation, Dmev is the mass equivalent diameter. The authors do not explain how the value of this mass equivalent diameter is obtained. According to the instruments used in this paper, it seems impossible to obtain Dmev. I am confused that this paper has calculated dynamic shape

factor by the value of Dmev.

---

## Author Comment (AC1) · 25 Oct 2019

The attachment includes: the revised manuscript, the revised supplementary, the reply to the Anonymous Referee. Please refer to the attachment for specific reply. (line 680-765).

Please also note the supplement to this comment: https://www.atmos-chem-phys-discuss.net/acp-2019-526/acp-2019-526-AC1-supplement.pdf

2019.

---

## Author Comment (AC2) · 25 Oct 2019

**Effective densities of soot particles and their relationships with the mixing state at an urban site of the Beijing mega-city in the winter of 2018**

Hang LIU1,2, Xiaole PAN1, Yu WU3, Dawei Wang1, Yu TIAN1,2, Xiaoyong LIU1,4, Lu LEI1,2, Yele 5 SUN1,2,4, Pingqing FU5, Zifa WANG1,2,4

[revised manuscript text omitted]

---

## Author Comment (AC3) · 25 Oct 2019

[revised manuscript text omitted]

**Uncertainty analyze:**

The physical parameters directly measured by the tandem system is the mass of rBC-containing particle ($M_p$), the mass of rBC core ($M_{rBC}$) and the mobility diameter of rBC-containing particle ($D_{mob}$).

$M_p$ is selected by CPMA and its uncertainty is influenced by the voltage and rotate speed of CPMA (Olfert and Collings, 2005).

In practice, the uncertainty can be determined through setting the resolution ($R_m$) parameter of CPMA. CPMA can change the voltage and rotate speed automatically to meet the uncertainty which was ~10% during our experiment.

The uncertainty of $D_{mob}$ has been determined to be ~3% (Kinney et al., 1991). The uncertainty of $M_{rBC}$ has determined to be ~30% (Shiraiwa et al., 2008).

For particle density, from the equation

$$\rho_{eff} = \frac{6M_p}{\pi D_{mob}{}^3} \tag{1}$$

Applying the propagation of uncertainty gives:

$$\left(\frac{\varepsilon_\rho}{\rho}\right)^2 = \left(\frac{\varepsilon_{M_p}}{M_p}\right)^2 + 9\left(\frac{\varepsilon_{D_{mob}}}{D_{mob}}\right)^2$$

Then the uncertainty of $\rho$ was determined to be 13.5%.

For dynamic shape factor, from the equation

$$\chi = \frac{D_{mob} \times C_c(D_{mev})}{D_{mev} \times C_c(D_{mob})} \tag{2}$$

Applying the propagation of uncertainty gives:

$$\left(\frac{\varepsilon_\chi}{\chi}\right)^2 = \left(\frac{\varepsilon_{D_{mob}}}{D_{mob}}\right)^2 + \left(\frac{\varepsilon_{D_{mev}}}{D_{mev}}\right)^2 + 2\left(\frac{\varepsilon_{C_c}}{C_c}\right)^2 \tag{3}$$

the $\varepsilon_{Cc}/Cc$ is the same for all particle sizes and equals to 2.1%(Allen and Raabe, 1985). The $D_{mev}$ is derived from equation 4-5 and the $\varepsilon_{Dmev}/D_{mev}$ is calculated to be ~4% .

$$M_p = \frac{\pi}{6}(D_{mev}^3 - D_c^3) * \rho_{coat} + \frac{\pi}{6}D_c^3 * \rho_{rBC} \tag{4}$$

$$D_{mev} = \sqrt[3]{\frac{6}{\pi}\left(M_p - \left(1 - \frac{\rho_{coat}}{\rho_{rBC}}\right) * M_{rBC}\right)} \tag{5}$$

Then, the uncertainty of $\chi$ was determined to be 5.8%.

For void ratio,

$$R_{void} = 1 - \frac{D_{mev}^3}{D_{mob}^3} \tag{6}$$

$$\left(\frac{\varepsilon_{Rvoid}}{R_{void}}\right)^2 = 9\left(\frac{\varepsilon_{D_{mob}}}{D_{mob}}\right)^2 + 9\left(\frac{\varepsilon_{D_{mev}}}{D_{mev}}\right)^2 \tag{7}$$

Then, the uncertainty of $R_{void}$ was determined to be 19.6%.

For mass ratio ($M_R$)

$$M_R = \frac{M_p - M_{rBC}}{M_{rBC}} \tag{8}$$

$$\left(\frac{\varepsilon_{MR}}{M_R}\right)^2 = \left(\frac{\varepsilon_{M_p}}{M_p}\right)^2 + \left(\frac{\varepsilon_{M_{rBC}}}{M_{rBC}}\right)^2 \tag{9}$$

Then, the uncertainty of $M_R$ was determined to be 31.6%.

Allen, M. D., and Raabe, O. G.: Slip Correction Measurements of Spherical Solid Aerosol-Particles in an Improved Millikan Apparatus, Aerosol Science and Technology, 4, 269-286, Doi 10.1080/02786828508959055, 1985.

Bond, T. C., Doherty, S. J., Fahey, D., Forster, P., Berntsen, T., DeAngelo, B., Flanner, M., Ghan, S., Kärcher, B., and Koch, D.: Bounding the role of black carbon in the climate system: A scientific assessment, Journal of Geophysical Research: Atmospheres, 118, 5380-5552, 2013.

Kinney, P. D., Pui, D. Y. H., Mulholland, G. W., and Bryner, N. P.: Use of the Electrostatic Classification Method to Size 0.1 Mu-M Srm Particles - a Feasibility Study, J Res Natl Inst Stan, 96, 147-176, 10.6028/jres.096.006, 1991.

Olfert, J. S., and Collings, N.: New method for particle mass classification - the Couette centrifugal particle mass analyzer, J Aerosol Sci, 36, 1338-1352, 10.1016/j.jaerosci.2005.03.006, 2005.

Shiraiwa, M., Kondo, Y., Moteki, N., Takegawa, N., Sahu, L., Takami, A., Hatakeyama, S., Yonemura, S., and Blake, D.: Radiative impact of mixing state of black carbon aerosol in Asian outflow, Journal of Geophysical Research: Atmospheres, 113, 2008.

**Reply to the comments of anonymous reviewer #1 on manuscript**

**Entitled " Effective densities of soot particles and their relationships with the mixing state at an urban site of the Beijing mega-city in the winter of 2018"**

We appreciate very much for the insight comments and recommendations of the reviewer in improving this paper and our future research. Here, we will response to all the comments one by one as follows:

1. Measurement procedures (Section 2.2) Aerosol particles were measured in the six steps using DMA-CPMA tandem system. In analytical condition, it took 10 minutes for each step (whole scan needs one hour). What is the response to
achieve equilibrium in step change? Usually, data after equilibrium condition were used in the stepwise measurements to reduce analytical errors. In other words, data immediately after step change were not suitable for analysis.

Reply: Yes, what the reviewer concerned is exactly right. The residence time between CPMA and SP2 is calculated to be ~19 s. Thus, the first 19 s in each step is easily influenced by the aerosols in the last step. In actual data analyze in this study, to carefully ensure the reliability of the data, the data in first 1 minute but not just 19 s of each step is abandoned.

2. Refractive index (Section 2.3.2) Refractive index of 1.48 + 0i was used for measurements of optical diameter. What is aerosol components with refractive index of 1.48 + 0i? This information is helpful for readers who are not familiar with aerosol optical properties.

Reply: The refractive index of 1.48 + 0i is the representative refractive index of ambient aerosols (mixture of various
components) and always used in the optical diameter calculation (Subramanian et al., 2010;Liu et al., 2014). Taylor et al. (2015) used the relative contribution of organics, nitrate and sulfate measured by SP-AMS to calculate the time-dependent refractive index of ambient aerosols and got the value of 1.46-1.50 (mean 1.48). They reported this refractive index is dominant by organics which is just similar to our cases. We will add the reference of (Taylor et al., 2015) to the refractive index of 1.48.

Also, as mentioned in the manuscript (line 201-203), the refractive indices of major components are reported for readers who are not familiar with optical properties as a reference.

3. Line 177-180 Effective density of non-rBC with 0.8-1.2 g cm$^{-3}$ (brue, yellow, and red) was no 20 % during EPs 1 and 5. The density was 30-40 % during EPs 1 and 5 in Fig. 1.

Reply: Thanks for the reminding. We have checked the data, the density was 20-40 % during the EPs 1 and 5 in Fig. 1. We will changed the number in the next version. However, this change don't influence the result since 20-40 % is still significantly lower than ~70% in EPs 2 and 4.

4. Line 206-207 and Fig. 2 Figure 2 provides us very important and interesting knowledge on relation between aerosol size
and effective density. Before detail discussion, influence of analytical error should be discussed. It is true that variation of refractive index can influence optical diameter. However, analytical error or analytical precision in the tandem system needs to be taken into account.

Reply: A detailed uncertainty analyze was conducted in the supplementary.

For the optical size exhibited in Fig. 2. It's transformed from the scattering signal directly measured by SP2. The scatter of
the distribution intensity of scattering can cause ~4% uncertainty of the optical diameter. The DMA-CPMA tandem system can cause ~3% uncertainty of the mobility diameter. However, the different choosing of refractive indices can cause up to ~14% uncertainty of the optical diameter. Thus, we mainly discuss about the uncertainty caused by the refractive indices.

5. Line 247-250 Rvoid value (0.92) was shown in this sentence. In Figure 3, Rvoid of 0.92 seems to be too large. Is that typo?

Reply: Yes, it's just a typo. 0.92 is actually the value of Rno_void, and the value of Rvoid is 0.08. We will change the number in the next version. We appreciate the reviewer pointed out this mistake.

6. Line 307-309 As shown in the text, insufficient detection limit can lead to disturb identification of detection of bi-modal pattern. Also, low analytical resolution in density may be one of the reasons.

Reply: The low analytical resolution may be truly one of the reasons. We will change the sentence to "We speculate that the $\rho_{rBC}$ number distribution will exhibit a bimodal pattern if the detection limit is sufficiently low and the analytical resolution is sufficiently high. The left peak in the expected bimodal pattern corresponds to the rBC-containing particles with thin
coatings." (Line 287-289)

7. Line 322-326 This difference between Eps 2 &4 and Eps 1 & 5 is very interesting in comparison of ambient aerosols. Are there any differences in meteorological conditions, air mass history, and so on? What were important or key processes to engender the differences?

Reply: We also found the difference between EPs 2&4 and EPs 1&5 which is very interesting. The air mass history didn't show distinct differences as shown in Fig. S5. We suspected the different meteorological conditions during these episodes leading to the different formation processes of aerosols (more organics in EPs 2&4). However, we think the major purpose of this paper is to explore the relationship between density and rBC's mixing state. Thus, we would like to investigate deeper in the next study to figure out the relationship among meteorological conditions, aerosols components and effective densities.

8. Figures 5-7 Analytical errors (i.e., error bars) should be added in the plots.

Reply: For Fig. 5, what we want to exhibit is the number distribution of rBC-containing particles and non-rBC with different effective densities. We think an error bar may be not necessary for such number distribution figures.

For Fig. 6, there have been an error bar in the figure.

For Fig. 7, we have added a new error bar denotes the analytical errors of the measured scattering cross section by SP2.

Liu, D., Allan, J. D., Young, D. E., Coe, H., Beddows, D., Fleming, Z. L., Flynn, M. J., Gallagher, M. W., Harrison, R. M., Lee, J., Prevot, A. S. H., Taylor, J. W., Yin, J., Williams, P. I., and Zotter, P.: Size distribution, mixing state and source apportionment
of black carbon aerosol in London during wintertime, Atmospheric Chemistry and Physics, 14, 10061-10084, 10.5194/acp-14-10061-2014, 2014.
Subramanian, R., Kok, G. L., Baumgardner, D., Clarke, A., Shinozuka, Y., Campos, T. L., Heizer, C. G., Stephens, B. B., de Foy, B., Voss, P. B., and Zaveri, R. A.: Black carbon over Mexico: the effect of atmospheric transport on mixing state, mass absorption cross-section, and BC/CO ratios, Atmospheric Chemistry and Physics, 10, 219-237, 10.5194/acp-10-219-2010,
2010.
Taylor, J. W., Allan, J. D., Liu, D., Flynn, M., Weber, R., Zhang, X., Lefer, B. L., Grossberg, N., Flynn, J., and Coe, H.: Assessment of the sensitivity of core/shell parameters derived using the single-particle soot photometer to density and refractive index, Atmos Meas Tech, 8, 1701-1718, 10.5194/amt-8-1701-2015, 2015.

**Reply to the comments of anonymous reviewer #2 on manuscript Entitled " Effective densities of soot particles and their relationships with the mixing state at an urban site of the Beijing mega-city in the winter of 2018"**

We appreciate very much the insight comments and recommendations of the reviewer in improving this paper and our future research. Here, we will response to all the comments one by one as follows:

General comments:

While the authors are fully aware that rBC particles in ambient air range in their effective densities from as low as 0.3 g/cm$^3$ (see Page 8 Line 230 and references therein), they arbitrarily set the lowest limit of detection to 0.8 g/cm$^3$, thus giving up a sizable fraction of rBC particles in ambient air. This makes their conclusions truncated that need to be supplemented with extrapolation and to some extent speculations. Why did they do this? One might presume that by setting the mobility diameter to 240 nm they perhaps thought that native (freshly emitted) rBC particles having very low effective densities are not relevant in this range?

Reply:

The initial purpose of this experiment is to investigate the relationship between the morphology of rBC-containing particles with their mixing state. The rBC-containing particles with different effective densities are representative of rBC-containing particles with different morphology. Thus, in the experiment design, we focus on the rBC-containing particles with relatively large effective density which have the opportunity to transform from a fractal structure to spherical structure. However, we admit we ignore the rBC-containing particles with less effective density and cannot show a whole spectrum of effective density distribution which was not our initial purpose of our experiment. Besides, there is actually a balance which need to be considered in a DMA-CPMA-SP2 measurement. As shown in Figure. 1 below, the detect efficiency (denoted by SP2/CPC) was not 100% for small rBC-containing particles ($M_p$<4 fg). The lower detection limit of this study is effective density of 0.8 g/cm$^3$ corresponding to $M_p$=5.79 fg. A lower effective density bound would cause large bias due to the detection limit of SP2. One solution is to increase the mobility diameter selected by DMA. However, another problem exists, since rBC-containing particle mainly locates in the small size range. The increasing in size of rBC-containing particles will significantly decrease the number detected by SP2 which will cause a big problem in the data interpretation especially in clean episodes when the number concentration of rBC-containing particle was typically low. For these two reasons, we decided a detection limit of 0.8 g/cm$^3$ which may miss the fresh emitted rBC-containing particles.

After this experiment, we are also very interested in the whole spectrum of the effective density of rBC-containing particles. We have thought about some ideas about addressing the problems we mentioned above and will conduct another experiment mainly focus on the whole spectrum of effective density of rBC-containing particles in this winter.

Although some results are from speculations, we think it is reasonable. And some results are quite certain which can help to understand the properties of rBC. For example, we found rBC-containing particles will transform to a near spherical structure when $M_R>7$. Besides, different cases were captured during this experiment including polluted and clean episodes and different polluted type (EP 2&4 and EP 1&5). We think such data is also very precious since the results concluded from such data can be more common.

[Figure]

**Figure 1 SP2's detection efficiency of Aquadag.**

2) In spite of the fact that several important physical parameters used in this study are derived by subtraction and division of measured quantities that were obtained by fundamentally different techniques (e.g. equations 4 and 5), the authors pay little if any attention to (propagated) uncertainties that can be huge in these cases.

Reply: We will add the uncertainties analyze in the supplementary. The uncertainties of the major parameters used in this paper are discussed.

Detailed comments:

Page 2 Line 37 'glacier reduction' is an imprecise term. Reduction in what? length? volume? albedo? and how? Sea ice and snow albedo is also reduced and melt is affected directly (e.g. by albedo reduction) and indirectly (e.g. by affecting radiative balance over reflective surfaces by absorbing reflected radiation)

Reply: Thanks for the reminding of the reviewer. Since BC can influence the glacier in multiple ways such as albedo, volume through direct and indirect effect. We would change "glacier reduction" to "glacier" in the next version. Since this sentence is only aimed to provide a background and emphasize the importance of BC, we don't want to extend and discuss more.

Page 2 Line 38 Second most important warming agent. There is no consensus yet whether methane or rBC comes second. Reference outdated, please update references and modify the statement accordingly

Reply: we have changed the expression and reference to"rBC is considered to be one of the most important global warming
factors.(Bond et al., 2013)."

Page 2 Line 40 'Visibility degradation' is not a major effect of rBC, it is mostly due to scattering aerosols. It is not the
human respiratory system that is directly 'harmed', but soot has many adverse health effects (including cardiovascular
illnesses, cancer, and even brain damage)

Reply: We agree with the reviewer and change the expression as well as the reference to"Additionally, as a component of
PM$_{2.5}$ (particulate matter with an aerodynamic diameter less than 2.5 μm), rBC has an adverse environmental effect by
harming human health leading to respiratory and cardiovascular illnesses as well as cancer (Apte et al., 2015;Lelieveld et al.,
2015;Raaschou-Nielsen et al., 2013;Dominguez-Rodriguez et al., 2015)." (line 39-41)

Reply: Great thanks for the reviewer's reminds. We will add "heterogenous chemistry" separately in the next version. (line
43)

Reply: Thanks, we have changed the expression. (line 52-53)

Reply: The expression has been changed to"Investigation on the ρeff of rBC-containing particles using a DMA-CPMA-CPC
tandem system would be difficult because there are substantial non-rBC particles in the ambient atmosphere." (line 78-80)

Reply: It's a typo, we have changed "principal" to "principle". (line 100)

Reply: We have changed the expression to"Then, the rBC is heated to incandescence." (line 102-103)

Reply: Firstly, we only assume the core-shell structure to calculate the optical diameter of BC, while the refractive indices
are measured by previous research(Taylor et al., 2015;Moteki et al., 2010) but aren't from assumption. We will change the expression to reduce misleading. The new expression is "By assuming a core-shell structure and using the refractive indices determine by (Taylor et al., 2015), 1.48 for coating and 2.26-1.26i for rBC core, the $D_{opt}$ of the rBC-containing particles can also be calculated based on the Mie-scattering theory." (line 147-149)

Secondly, using the core-shell assumption is just a method to study the morphology of rBC-containing particles. As we detailed in line 195-197, we compared the optical diameter with the mobility diameter and found rBC-containing particles with less effective density tended to have a less optical diameter although they had the same mobility diameter. This phenomenon may be caused by more fractal structure of rBC-containing particles with less effective density.

Thirdly, in my opinion, the optical diameter is actually a representative of particles' scattering ability in this study. We can also use the core-shell model to calculate the scattering intensity of rBC-containing particle with $D_{mob}= 240$ nm to compare with the measured scattering intensity of rBC-containing particles with varied effective density. We suppose we will get the same result by doing so.

Finally, such a core-shell assumption is always used in previous research to transform scattering intensity to optical diameter of rBC-containing particles (Han et al., 2019;Zhang et al., 2018;Laborde et al., 2013). Our result can also suggest one can't simply regard optical diameter as the actual diameter for rBC-containing particles typically for fresh rBC-containing particles with less effective density.

Page 6 Line 165: uncertainty should be reported here since the two parameters are determined by two principally different methods having their own inherent uncertainties

Reply: We add the uncertainty of $M_R$ in line 165-166 and the uncertainties of the major parameters using in this paper can be found in supplementary.

Page 6 Line 171-172 'microgram/cm3' ?

Reply: We think " $\mu g/cm^3$" have been widely used in previous research and we want to keep consistent with previous research.

Page 6 Line 175 'density' use plural

Reply: Thanks, we have changed the expression.

Page 7 Line 195 this statement is definitely not true. Bulk non-rBC particles should differ from the coatings of rBC due to differences in chemistry of their formations.

Reply: As we can see from Fig. S6, the effective densities of bulk non-rBC tended to be less at the condition when there was more organic fraction in the NR-PM$_{2.5}$. We think, the bulk effective density of non-rBC can reflect the organic fraction in non-rBC particles since the densities of organic compounds tend to be less than that of inorganic compounds.

Then, the bulk effective of rBC-containing particles also tended to be less at the condition when there was more organic fraction in the NR-PM$_{2.5}$ which may be the result of more organic coating.

    Thus, our logic is the less bulk non-rBC effective density represents the more organic fraction condition leading to more organic coating of rBC.

    We agree the bulk non-rBC particles should differ from the coatings of rBC due to differences in chemistry of their formations. However, the simultaneous decrease of the bulk effective densities of non-rBC and rBC-containing particles when organic fraction increased (Fig. S6) suggests there was some relationship between the composition of non-rBC and the coating of rBC-containing particles. Thus, the bulk effective density of non-rBC can reflect the composition of coatings of rBC in some degree.

    For preciseness, we will change "same" to "similar" in the manuscript. (line 195)

Page 8 Line 243-249 The whole statement is highly speculative (see my comment immediately above)

    Reply: We have carefully considered the comments of the reviewer. We agree this statement lacks direct evidence, such as the direct measurement of the composition of non-rBC and rBC-containing particles to support the argument. However, we still believe the effective density can reflect the composition of non-rBC and the coating of rBC in some degree. Due to instrument limits, we are not able to do more experiments to approve this assumption at present. We would like to explore such relationship further in the future.

    In this paper, we will weaken the statement about the coating composition effects on rBC. Line 243-249 has been rewritten (now line 240-245).

    The former statement: "Since the $\rho_{rBC}$ was influenced by the combined effect of the coating chemical composition and morphology, the variation in $\chi$ and R$_{void}$ were separately counted in different $\rho_{non\text{-}rBC,bulk}$ situations representing the different coating composition conditions. The rBC-containing particles with a lower coating effective density ($1.1 < \rho_{non\text{-}rBC,bulk} < 1.3$ g/cm$^3$) could reach a compact spherical structure when the $\rho_{rBC}$ was 1.2 g/cm$^3$ with an $\chi$ of 1.05 and a R$_{void}$ of 0.92, whereas the morphological transition of $\rho_{rBC}$ was higher for rBC-containing particles at a higher $\rho_{non\text{-}rBC,bulk}$ condition."

    The new version: We found that rBC-containing particle had a larger $\chi$ value and R$_{void}$ at the condition when $\rho_{non\text{-}rBC}$,bulk is smaller, especially for irregular particles (Fig. 3a). It may imply that different coating composition played a different role in the morphology reconstructing of rBC-containing because $\rho_{non\text{-}rBC,bulk}$ reflected the composition of non-rBC which may relate to the coating composition of rBC in some degree (Fig. S6). The rBC-containing particles could reach a compact spherical structure when the $\rho_{rBC}$ was 1.2 g/cm3 with an $\chi$ of 1.05 and a R$_{void}$ of 0.08 when 1.1 g/cm3 $< \rho_{non\text{-}rBC,bulk} <$ 1.3 g/cm3 , whereas the morphological transition of $\rho_{rBC}$ was higher for rBC-containing particles at a higher $\rho_{non\text{-}rBC,bulk}$ condition.

The former line 270-283 has been deleted.

Reply: The initial purpose of this experiment is to investigate the relationship between the morphology of rBC-containing particles with their mixing state. The rBC-containing particles with different effective densities are representative of rBC-containing particles with different morphology. Thus, in the experiment design, we focus on the rBC-containing particles with relatively large effective density which have the opportunity to transform from a fractal structure to spherical structure.

However, we admit we ignore the rBC-containing particles with less effective density and cannot show a whole spectrum of effective density distribution which was not our initial purpose of our experiment. Besides, there is actually a balance which need to be considered in a DMA-CPMA-SP2 measurement. As shown in Figure. 1 below, the detect efficiency (denoted by SP2/CPC) was not 100% for small rBC-containing particles ($M_p < 4$ fg). The lower detection limit of this study is effective density of 0.8 g/cm$^3$ corresponding to $M_p = 5.79$ fg. A lower effective density bound would cause large bias due to the detection limit of SP2. One solution is to increase the mobility diameter selected by DMA. However, another problem exists, since rBC-containing particle mainly locates in the small size range. The increasing in size of rBC-containing particles will significantly decrease the number detected by SP2 which will cause a big problem in the data interpretation especially in clean episodes when the number concentration of rBC-containing particle was typically low. For these two reasons, we decided a detection limit of 0.8 g/cm$^3$ which may miss the fresh emitted rBC-containing particles.

After this experiment, we are also very interested in the whole spectrum of the effective density of rBC-containing particles. We have thought about some ideas about addressing the problems we mentioned above and will conduct another experiment mainly focus on the whole spectrum of effective density of rBC-containing particles in this winter.

Although some results are from speculations, we think it is reasonable. And some results are quite certain which can help to understand the properties of rBC. For example, we found rBC-containing particles will transform to a near spherical structure when $M_R > 7$. Besides, different cases were captured during this experiment including polluted and clean episodes and different polluted type (EP 2&4 and EP 1&5). We think such data is also very precious since the results concluded from such data can be more common.

2. Comment on the calculation and definition of effective density This paper presents two methods of effective densities calculation, but I do not quite understand them. Previous studies (Qiao et al., 2018; Momenimovahed and Olfert, 2015) generally used a log-normal or Gaussian function to fit the eff distribution. The eff of the bulk aerosols was determined to be the peak location of the fit function. In this paper, however, the first calculation method (Lines 125-130) is, "Particles with known effective densities preselected by the DMA-CPMA system were injected into the SP2 to obtain information on the corresponding BC. In practice, the mobility diameter selected by the DMA was set at a constant value of 240 nm. The setpoints of the CPMA were 5.79, 7.24, 8.69, 10.13, 11.58, and 13.03 fg, which corresponded to a eff of 0.8,1.0, 1.2, 1.4, 1.6, and 1.8 g/cm3, respectively." This calculation is totally different from the calculation in the previous studies. Is no a log-normal or Gaussian function fitted? It is imprecise in theory and in practice. What is the strategy about the selection of CPMA data? The second calculation method defines a new effective density which names the bulk aerosol density, as stated in Lines 137. In my opinion, it should be the bulk aerosol effective density. Additionally, the authors simply use the PSL to demonstrate this method, which lacks the experiments about rBC.

Reply: We want to emphasize that there are two effective densities in this paper, one is the effective density of a single particle and the other is the effective density of aerosol bulk since aerosol bulk has various aerosols with different effective densities.

The previous studies which we listed is just the method to determine the aerosol bulk effective density and can't be simply compared with the method we select particles with a certain effective density. As a matter of fact, the calculation and the theory of previous research are just similar to our study. What they do is to measure more effective densities points. For example, "the mobility diameter selected by the DMA was set at a constant value of 240 nm. The setpoints of the CPMA were …,5.79,…, 7.24,…, 8.69,…,10.13,…,11.58,…,and,…,13.03fg, which corresponded to an effective density of 0.5,0.6,0.7,0.8,0.9,1.0,1.1,1.2,1.3,1.4,1.5,1.6,1.7,1.8 g/cm$^3$, respectively" in their paper. Thus, with more effective density points, one can fit a log-normal or Gaussian function to determine the peak location of the effective density distribution. The main purpose of such fitting is to find the effective density with the maximum number counts to represent the effective density characteristic of the bulk aerosol. In our study, due to the detection resolution, we adopt a new method whose main purpose is also to find effective density of the maximum number counts. As long as the number counts of each effective density is measured correctly, this method will not cause an uncertainty larger than 0.1 g/cm$^3$ in the determination of the eff of the bulk effective density in our study.

3. Comment on the setup of DMA-CPMA-SP2 This study uses a novel setup to characterize the effective density of rBC particles in the atmosphere, but this setup does not be verified by rBC particles produced in laboratory. I strongly suggest that it should be assessed before applying it to the field observation.

Reply: Yes, this is a good advice and we would like to do so to make our result more convincing. However, there is some trouble with our SP2's flow system now and it can't operate normally.

However, I can give you an evidence in another aspect. We measured the effective density of Aquadag using the DMA-CPMA-CPC system in previous study which showed a good consistency with other researchers (Figure. 1). In DMA-CPMA-SP2 system, we can say surely we can get the same result as DMA-CPMA-CPC system in our previous study, since the function of SP2 is to detect the number of rBC which shows a good consistency with CPC (Figure. 2).

[Figure]

Figure 1 Relationship between effective density and mobility diameter of Aquadag measured by DMA-CPMA-CPC system and in previous studies.

[Figure]

Figure 2 SP2's detection efficiency of Aquadag.

4. Comment on the determination of the shape factor In 2.3.3, author uses the equation (3) to determine the dynamic shape factor. In the equation, $D_{mev}$ is the mass equivalent diameter. The authors do not explain how the value of this mass equivalent diameter is obtained. According to the instruments used in this paper, it seems impossible to obtain $D_{mev}$. I am confused that this paper has calculated dynamic shape factor by the value of $D_{mev}$.

Reply: The $D_{mev}$ is calculated by solving the following equation:

$$M_{coat} = (\frac{1}{6} * \pi * D_{mev}^3 - \frac{1}{6} * \pi * D_c^3) * \rho_{coat} = M_p - M_{rBC}$$

$\rho_{coat}$ is assumed to be the same as $\rho_{non-rBC,bulk}$ in this study. (see the response to the anonymous reviewer #2)